# Proteomic Analysis of HCC-1954 and MCF-7 Cell Lines Highlights Crosstalk between αv and β1 Integrins, E-Cadherin and HER-2

**DOI:** 10.3390/ijms231710194

**Published:** 2022-09-05

**Authors:** Denise de Abreu Pereira, Vanessa Sandim, Thais F. B. Fernandes, Vitor Hugo Almeida, Murilo Ramos Rocha, Ronaldo J. F. C. do Amaral, Maria Isabel D. Rossi, Dário Eluan Kalume, Russolina B. Zingali

**Affiliations:** 1Programa de Oncobiologia Celular e Molecular (POCM), Coordenação de Pesquisa, Instituto Nacional do Câncer, Rio de Janeiro 20231-050, Brazil; 2Unidade de Espectrometria de Massas e Proteômica (UEMP), Instituto de Bioquímica Médica Leopoldo de Meis, Universidade Federal do Rio de Janeiro, Rio de Janeiro 21941-902, Brazil; 3Laboratório de Hemostase e Venenos (LABHEMOVEN), Instituto de Bioquímica Médica Leopoldo de Meis, Universidade Federal do Rio de Janeiro, Rio de Janeiro 21941-902, Brazil; 4Instituto de Bioquímica Médica Leopoldo de Meis, Universidade Federal do Rio de Janeiro, Rio de Janeiro 21941-902, Brazil; 5Laboratório de Proliferação e Diferenciação Celular, Instituto de Ciências Biomédicas, Universidade Federal do Rio de Janeiro, Rio de Janeiro 21941-902, Brazil; 6Instituto de Ciências Biomédicas e Hospital Universitário Clementino Fraga Filho, Universidade Federal do Rio de Janeiro, Rio de Janeiro 21941-617, Brazil; 7Laboratório Interdisciplinar de Pesquisas Médicas (LIPMed), Instituto Oswaldo Cruz, Fundação Oswaldo Cruz, Rio de Janeiro 21040-360, Brazil

**Keywords:** breast cancer, HER-2, MCF-7, HCC-1954, proteome, membrane proteins

## Abstract

Overexpression of human epidermal growth factor receptor-2 (HER-2) occurs in 20% of all breast cancer subtypes, especially those that present the worst prognostic outcome through a very invasive and aggressive tumour. HCC-1954 (HER-2+) is a highly invasive, metastatic cell line, whereas MCF-7 is mildly aggressive and non-invasive. We investigated membrane proteins from both cell lines that could have a pivotal biological significance in metastasis. Membrane protein enrichment for HCC-1954 and MCF-7 proteomic analysis was performed. The samples were analysed and quantified by mass spectrometry. High abundance membrane proteins were confirmed by Western blot, immunofluorescence, and flow cytometry. Protein interaction prediction and correlations with the Cancer Genome Atlas (TCGA) patient data were conducted by bioinformatic analysis. In addition, β1 integrin expression was analysed by Western blot in cells upon trastuzumab treatment. The comparison between HCC-1954 and MCF-7 membrane-enriched proteins revealed that proteins involved in cytoskeleton organisation, such as HER-2, αv and β1 integrins, E-cadherin, and CD166 were more abundant in HCC-1954. β1 integrin membrane expression was higher in the HCC-1954 cell line resistant after trastuzumab treatment. TCGA data analysis showed a trend toward a positive correlation between HER-2 and β1 integrin in HER-2+ breast cancer patients. Differences in protein profile and abundance reflected distinctive capabilities for aggressiveness and invasiveness between HCC-1954 and MCF-7 cell line phenotypes. The higher membrane β1 integrin expression after trastuzumab treatment in the HCC-1954 cell line emphasised the need for investigating the contribution of β1 integrin modulation and its effect on the mechanism of trastuzumab resistance.

## 1. Introduction

Breast cancer is the most frequent tumour and a leading cause of death among women worldwide [1]. Increasing our understanding of the biological mechanisms involved in breast cancer development, implementation of screening programs for early detection of the disease, and new advances in treatment using combined therapies have contributed to a decline in mortality rates [2]. As a heterogeneous disease, breast cancer has different responses to therapeutic treatment based on the specific molecular gene expression and biochemical profile. It can be classified as luminal A, which has a low proliferative rate (measured by Ki67 protein expression) and is positive for estrogen receptor (ER) and/or progesterone receptor (PR); luminal B, which has a high Ki67 expression and is ER and/or PR positive; HER-2 positive, which has a high level of human epidermal growth factor receptor HER-2 amplification/expression; or a heterogenous group of tumours characterised by the absence of ER, PR, and HER-2 that are classified as triple-negative breast cancer (TNBC) [3,4,5].

Overexpression of HER-2/ErbB2 occurs in 20% of all breast cancer subtypes, and it can control processes such as cellular growth, apoptosis, proliferation, differentiation, angiogenesis, and invasion by regulating PI3K/AKT and MAPK/ERK signalling pathways [6]. HER-2 positive breast cancer patients present the worst prognostic outcome, characterised by a very invasive and aggressive tumour that generally develops brain metastasis [7]. In the early 2000s, the introduction of a targeted therapy with trastuzumab (a humanised monoclonal antibody) reduced recurrence and mortality of HER-2 positive breast cancer patients [2,8,9]. However, some patients develop resistance to the treatments, leading to the appearance of a more aggressive disease with the development of metastasis [10,11,12]. The metastatic process involves loss of adhesion, invasion of the extracellular matrix, intravasation into the blood and/or lymphatic system, extravasation into the microvasculature of distant organs, colonisation, and proliferation of the secondary tumour at the metastatic site [13].

Membrane proteins have a pivotal biological significance in all these steps of the metastatic process. The cadherin family, for example, plays an important role in mediating cell-to-cell adhesion in breast cancer metastasis [14]. The adherence of tumour cells to the extracellular matrix (ECM) is mediated by integrins, which are transmembrane receptors that interact with ECM components such as fibronectin, laminin, collagen, fibrinogen, and vitronectin [15]. Integrins are also known to participate in the modulation of tumour migration potential by activation of metalloproteases [16,17]. Therefore, the proteomic evaluation of differential abundance of membrane proteins could be of value for understanding the differences in breast tumours.

The cell lines HCC-1954 and MCF-7 represent breast cancer subtypes of different aggressiveness. HCC-1954 is a poorly differentiated breast cancer cell line isolated from a patient with a grade 3 IIA primary invasive ductal carcinoma that overexpresses HER-2, with no lymph node metastases. It is negative for ER/PR receptors and is resistant to trastuzumab treatment [18]. This cell line has been the subject of several genomic analyses that detected somatic point mutations and chromosome translocations [19]. Some translocations that generated chimeric genes were identified, validated, and associated with cell phenotypes related to cancer development and progression [20]. MCF-7 was isolated from the pleural effusion of a 69-year-old woman with metastatic disease. It is considered to be a poorly aggressive and non-invasive cell line that has low metastatic potential and retains several characteristics of differentiated mammary epithelium, including the capability of forming domes [21].

In this work, we carried out a label-free quantitative proteomic comparison of membrane proteins from HCC-1954 and MCF-7 cell lines that provides insight into the biological differences between a highly invasive and metastatic cell and a non-invasive cell, respectively. For this purpose, we enriched membrane proteins from both cell lines using a biotin approach, a filter-aided sample preparation (FASP) method to enable in-solution trypsin digestion, and then we performed quantitative label-free 2D-LC/MS proteomic analysis of the membrane-enriched (ME) fraction and the flow-through (FT) for both HCC-1954 and MCF-7 breast cancer cell lines. Differentially expressed proteins that have been implicated in adhesion, invasion, and dissemination of cancer cells were selected and validated by flow cytometry, Western blot, and immunohistochemical analysis. In addition, β1 integrin expression was analysed by Western blot in cells upon trastuzumab treatment.

## 2. Results

### 2.1. Analysis of HER-2 Expression in HCC-1954 and MCF-7 Cell Lines Confirm the Difference of Their Breast Cancer Phenotypes

Immunofluorescence analysis (Figure 1A) showed a staining pattern of HER-2 on the membrane and in the cytoplasm of HCC-1954 cells as previously observed by other studies [22,23]. As expected, no HER-2 was detected by immunofluorescence in MCF-7 cells. HER-2 was observed in the membrane-enriched fraction and in the FT fraction of HCC-1954, whereas in MCF-7 fractions HER-2 was not detected (see Western blot analysis in Figure 1B and proteomic data analysis in Appendix A). These results confirmed the difference of HCC-1954 and MCF-7 breast cancer phenotypes.

### 2.2. Label-Free LC-MS^E^ Proteomic Data Quality Analysis Shows Confidence in Protein Expression among the Different Samples

Data quality analyses (Appendix A) indicated that the hydrolysis process was successful and that reasonable ionisation and fragmentation were achieved. Power analysis showed that approximately 90% of our data had a power >0.8 using four replicates, indicating that the probability of the data being true was over 80% and the number of replicates used for the experiment was truly sufficient (Appendix A). The principal component analysis (PCA) showed the distribution of the proteins identified and quantified among the four experimental replicates of each sample. It was observed from the two-dimensional plane that the four samples occurred in four distinct clusters corresponding to the four replicates. This suggested a good reproducibility of the technical replicates and the injections, reflecting a quantitative difference in the expression of the proteins distributed among the different samples (Appendix A).

### 2.3. HCC-1954 and MCF-7 Label-Free LC-MS^E^ Proteomic Statistical Analysis Filtered a Total of 450 Proteins

A total of 1386 proteins were identified and quantified by label-free LC-MSE, 311 in the HCC-1954 ME fraction, 403 in the HCC-1954 FT fraction, 276 in the MCF-7 ME fraction and 396 in MCF-7 FT fractions. After statistical analysis a total of 450 proteins from ME and FT fractions from HCC-1954 and MCF-7 were filtered with a 95% confidence level (ANOVA *p* < 0.05) by the Progenesis QI analysis (Appendix A, Appendix A). Forty-two proteins were present in only one fraction: in HCC-1954 there were 4 in ME and 15 in FT, and in MCF-7 there were 10 in ME and 13 in FT. Twenty-eight proteins were present only in the HCC-1954 cell line and 25 proteins only in the MCF-7 cell line. Nine out of 28 proteins from HCC-1954 were found in both ME and FT fractions, while 2 out of 25 proteins from MCF-7 were present in both fractions (Appendix A).

We focused our analysis on membrane-enriched proteins from both cell lines; a total of 343 proteins were identified and quantified between ME fractions. Figure 2A shows a Venn diagram with the protein distribution of membrane-enriched fractions from HCC-1954 and MCF-7 cell lines. A total of 244 proteins were common to HCC-1954 and MCF-7 ME fractions; 67 were identified only in HCC-1954 while 32 were found only in MCF-7 when comparing ME fractions.

### 2.4. Proteins Identified and Quantified for HCC-1954 and MCF-7 Cell Line Fractions Show Statistically Significant Differences in Abundance

To analyse the statistical significance of the differences in abundance among all proteins identified and quantified in HCC-1954 and MCF-7, volcano plots were generated separately for ME or FT fractions (Figure 2B and Appendix A), respectively.

The volcano plot for the ME fraction showed that a total of 155 proteins reflecting the differences in the ME fraction for the ratio (HCC-1954/MCF-7) were statistically significant for false discovery ratio (FDR) = 0.01; s0 = 0.5, and *p* < 0.05. Of these, 82 proteins showed increased (Figure 2B, black circles) and 73 proteins showed decreased (Figure 2B, grey circles) relative fold change (FC). This was the relative protein abundance measured by the HCC-1954/MCF-7 ratio. Additionally, we first analysed ME/FT separated for each cell line and then compared proteins from ME fractions. A total of 103 proteins were statistically significant, of which 76 showed increased and 27 decreased relative FC (Table 1).

Regarding the flow-through fractions (FT), 88 proteins were statistically significant with FDR = 0.05; s0 = 0.2, and *p* < 0.05, reflecting the differences in FT fractions for the HCC-1954/MCF-7 ratio. Of these, 46 proteins showed increased (Appendix A, black circles) and 42 decreased (Appendix A, grey circles) relative fold change. Appendix A show the heat map analyses of ME and FT fractions, respectively.

**Table 1 ijms-23-10194-t001:** Proteins from ME fractions that were relatively increased when compared to FT fractions.

HCC-1954	MCF-7
14-3-3 protein theta YWHAQ	26S proteasome regulatory subunit 6A PSMC3
3-hydroxyacyl-CoA dehydrogenase type-2 HSD17B10	4-aminobutyrate aminotransferase, mitochondrial ABAT
40S ribosomal protein S19 RPS19	Acyl-CoA dehydrogenase family member 9, ACAD9
40S ribosomal protein S27 RPS27	Alpha-1-antichymotrypsin SERPINA3
45 kDa calcium-binding protein SDF4	Cell division cycle and apoptosis regulator protein 1 CCAR1
6-phosphogluconate dehydrogenase, decarboxylating PGD	Core histone macro-H2A.1 H2AFY
Acetyl-CoA acetyltransferase, mitochondrial ACAT1	DNA-(apurinic or apyrimidinic site) lyase APEX1
Actin, aortic smooth muscle ACTA2	Elongation factor 1-delta EEF1D
Actin, cytoplasmic 2 ACTG1	Enoyl-CoA hydratase, mitochondrial ECHS1
Actin-related protein 2/3 complex subunit 1B ARPC1B	Eukaryotic translation initiation factor 2 subunit 3 EIF2S3
Adenosylhomocysteinase AHCY	Eukaryotic translation initiation factor 4B EIF4B
ADP,ATP carrier protein, heart isoform T1 SLC25A4	Heterogeneous nuclear ribonucleoprotein A3 HNRNPA3
Aldehyde dehydrogenase family 1 member A3 ALDH1A3	Histone H1.2 HIST1H1C
Alpha-actinin-1 ACTN1	Histone H2A.V H2AFV
Alpha-enolase ENO1	Kininogen-1 KNG1 PE = 1 SV = 2
Beta-2-microglobulin B2M	Leucine--tRNA ligase, cytoplasmic LARS
Cadherin-1 CDH1	Methyl-CpG-binding protein 2 MECP2
Catenin alpha-1 CTNNA1	Nectin-1 NECTIN1
Catenin delta-1 CTNND1	Neuroblast differentiation-associated protein AHNAK
CD166 antigen ALCAM	Phosphoenolpyruvate carboxykinase [GTP], PCK2
CD44 antigen CD44	Pleckstrin homology domain-containing family A member 4 PLEKHA4
CD59 glycoprotein CD59	Pregnancy-specific beta-1-glycoprotein 6 PSG6
Coiled-coil domain-containing protein 170 CCDC170	Pyruvate carboxylase, mitochondrial PC
Core histone macro-H2A.2 H2AFY2	Septin-11 SEPTIN11
Cytochrome b5 reductase 4 CYB5R4	Splicing regulatory glutamine/lysine-rich protein 1 SREK1
Cytochrome P450 4F11 CYP4F11	START domain-containing protein 10 STARD10
D-3-phosphoglycerate dehydrogenase PHGDH	Twinfilin-1 TWF1
DnaJ homolog subfamily C member 2 DNAJC2	
Elongation factor 2 EEF2	
Far upstream element-binding protein 3 FUBP3	
Fatty acid synthase FASN	
Filamin-A FLNA	
Filamin-B FLNB	
Fructose-bisphosphate aldolase C ALDOC	
Galectin-1 LGALS1	
Heat shock 70 kDa protein 1 (HSP70-1)	
Hepatoma-derived growth factor HDGF	
Inactive caspase-12 CASP12	
Integrin alpha-V ITGAV	
Integrin beta-1 ITGB1	
Keratin, type I cytoskeletal 15 KRT15	
Keratin, type I cytoskeletal 18 KRT18	
Keratin, type II cytoskeletal 6B KRT6B	
Keratin, type II cytoskeletal 7 KRT7	
Keratin, type II cytoskeletal 8 KRT8	
Kunitz-type protease inhibitor 1 SPINT1	
Metalloproteinase inhibitor 2 TIMP2	
Mitochondrial import receptor subunit TOM70 TOMM70	
MORN repeat-containing protein 1 MORN1	
Myosin light polypeptide 6 MYL6	
Myosin-10 MYH10	
Myosin-9 MYH9	
NKAP-like protein NKAPL	
Nucleolin NCL	
Nucleoside diphosphate kinase A NME1	
Pachytene checkpoint protein 2 homolog TRIP13	
POTE ankyrin domain family member F POTEF	
Protein FAM118B FAM118B	
Protein-glutamine gamma-glutamyltransferase 2 TGM2	
Putative Ras-related protein Rab-1C RAB1C	
Receptor tyrosine-protein kinase erbB-2 ERBB2	
Serine/threonine-protein phosphatase 2A 65 kDa regulatory subunit A alpha isoform PPP2R1A	
Short-chain dehydrogenase/reductase 3 DHRS3	
Sulfide:quinone oxidoreductase, mitochondrial SQOR	
T-box brain protein 1 TBR1	
T-complex protein 1 subunit alpha TCP1	
T-complex protein 1 subunit delta CCT4	
T-complex protein 1 subunit epsilon CCT5	
T-complex protein 1 subunit theta CCT8	
T-complex protein 1 subunit zeta CCT6A	
Tripartite motif-containing protein 77 TRIM77	
Tropomyosin alpha-3 chain TPM3	
Tropomyosin alpha-4 chain TPM4	
UTP–glucose-1-phosphate uridylyltransferase UGP2	
Y-box-binding protein 3 YBX3	
ZAR1-like protein ZAR1L	

### 2.5. Gene Ontology Analysis of Proteins Upregulated in HCC-1954 or MCF-7 ME Fractions Indicates the Cell Line Characteristics That Account for Their Phenotype Differences

To understand the biological significance of the proteins detected by proteomic analysis, we selected up- or down regulated proteins with a relative fold change greater than 2 (0.5 < FC > 2) for the HCC-1954/MCF-7 ratio of ME fractions. This resulted in 73 proteins upregulated in HCC-1954, whereas 69 proteins were upregulated in MCF-7, which might account for differences in the phenotypes and aggressiveness of these cell lines. Then we analysed the GO classification of these proteins using the Panther, Proteomaps, and KEGG platforms (Appendix A, sheets 10 to 13). The Panther analysis (Figure 2C, D) transforms the protein ID in the gene ID to perform the gene ontology analysis. The gene hits correspond to how many times proteins related to that gene appear. We identified proteins classified as cytoskeletal proteins, scaffold/adaptor proteins, cell adhesion molecules, membrane traffic proteins, protein binding activity modulators, and calcium-binding proteins in HCC-1954 (Figure 2C). Proteins from MCF-7 were classified as proteins for chromatin/chromatin binding and regulatory proteins, translational proteins, and nucleic acid metabolism proteins (Figure 2C). The pathway classifications of these proteins in HCC-1954 or MCF-7 breast cancer cells were quite different. Proteins from HCC-1954 were classified as EGF receptor pathway, cytoskeletal regulation by RHO GTPase, nicotinic acetylcholine receptor signalling pathway, cadherin signalling pathway, and FGF signalling pathway (Figure 2D), whereas proteins from MCF-7 were classified as p53 pathway, ATP synthesis, p38 MAPK pathway, angiogenesis and DNA replication (Figure 2D).

Gene ontology analysis also indicated that some general pathways of metabolism and homeostasis, such as metabolite interconversion enzyme, nucleic acid metabolism protein, DNA replication, pentose phosphate pathway, and ATP synthesis, were activated in both cell lines. This indicated that both cells functioned to evade growth cell suppression, to sustain proliferation, to generate genomic instability and mutation. All these processes are classified as cancer hallmarks and reflect the tumours cells’ auto sufficiency [24].

### 2.6. Highly Expressed Proteins of HCC-1954 Could Be Associated with Cell Mechanobiology

We used proteomaps, in which the polygon size was proportional to the protein abundance, and proteins functionally related were arranged together to highlight how cells are working based on the protein abundances and functions. In HCC-1954 the main biological processes were classified as tight junctions and cytoskeleton proteins (Figure 3A). The highly abundant proteins associated with these processes were MYH9 (FC = 14) and ACTG1 (FC = 2.5) (Figure 3B). Conversely, in MCF-7 two genetic information processes stood out involving proteasomes and histones (Figure 3C). The protein PSMC3 (FC = 2.2) was the major protein functionally related to the proteasome pathway (Figure 3D), and the proteins H2AFY (FC = 7), H2AFV (FC = 10), HIST2H4B (FC = 8), and HIST2H2BF (FC = 17) were associated with histone processing (Figure 3D).

Additionally, Figure 3E shows the top pathways analysed using KEGG. It is apparent that HCC-1954 had more proteins involved with tight junctions and the Hippo signalling pathway, regulation of the actin cytoskeleton, and adherens junctions, while MCF-7 had proteins related to neutrophil extracellular trap formation, systemic lupus erythematous, and viral carcinogenesis. The proteins of the HCC-1954 and MCF-7 cell lines classified in the top pathways are shown in Appendix A, respectively. Most of these proteins were classified in more than one pathway, and all these pathways can be related to cytoskeleton structure and function, as they contribute to cell shape, motility, proliferation, and division.

We used proteins from the HCC-1954 or MCF-7 ME fractions with an increased abundance (FC > 2) to create the protein–protein interaction network using the STRING platform. In the HCC-1954 cell line most of these proteins produced only one group of interactions in which all the proteins showed an association with more than one other protein (Figure 3F), predominantly interlinked among the processes of cytoskeleton organisation and function. The MCF-7 cell line protein network suggested four nodes of interaction that were interconnected (Figure 3G). They were linked to cellular component organisation, epigenetic regulation of gene expression, methylation of chromatin, and metabolic processes.

### 2.7. Integrin β1 and Cadherin-1 Expression in HCC-1954 and MCF-7 Cell Lines Are in Accordance with Cells’ Characteristic Phenotype

The proteomic analyses revealed a higher expression of β1 integrin (FC = 2.24) and E-cadherin (FC = 13.39) in HCC-1954 than in MCF-7 in the ME fraction. Panther analysis showed an increase in proteins related to cadherin and integrin signalling pathways (Figure 2C). Proteomaps in the upper left corner of Figure 3B show that catenin delta-1 (CTNND1), cadherin-1 (CDH1), and integrin beta-1 (ITGB1) were the highly functionally related proteins classified in the Rap1 signalling pathway. Rap1 signalling pathway activity is involved in cell adhesion, migration, and invasion [25,26]. The STRING network showed a higher degree of interactions between HER-2, cadherin-1, and integrins in HCC-1954 ME (Figure 3F). To confirm the differences in the expression of cadherin-1 and integrins in HCC-1954 and MCF-7 cell lines, we used flow cytometry, immunofluorescence, and Western blot analyses.

Figure 4A–H shows the membrane expression of β1 (CD29), α2 (CD49-b), αv (CD51), and β3 (CD61) integrins in HCC-1954 and MCF-7 cells, respectively. A higher expression of integrin β1 (CD29) and a low expression of integrin β-3 were observed in HCC-1954. Integrin β3 (CD61) was not detected in MCF-7 cells. Expression of α2 (CD49b), αv (CD51), and β1 (CD29) integrins was greater in HCC-1954 than in MCF-7 cells, as indicated by their mean fluorescent intensity (MFI) (Figure 4I).

We confirmed E-cadherin expression in HCC-1954 and MCF-7 cells by immunofluorescence (Figure 4J). HCC-1954 showed a higher membrane and cytosol immunofluorescence than MCF-7 cells. Figure 4K,L shows the Western blot analysis and quantification of total extracts (TEs), membrane-enriched (ME), and flow-through (FT) fractions of HCC-1954 and MCF-7 cells. In HCC-1954, E-cadherin staining was more intense in the ME fraction, whereas in MCF-7 cells it was the FT fraction that stained more intensely.

**Figure 4 ijms-23-10194-f004:**
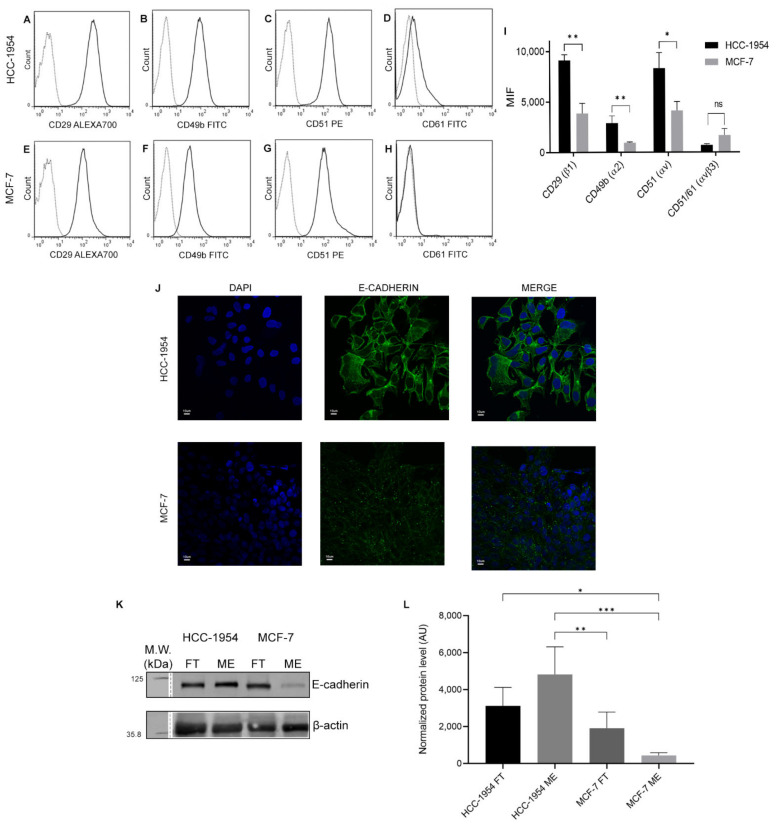
Integrin phenotypes and E-cadherin protein expression of HCC-1954 and MCF-7 human breast cancer cell lines. Histograms shows the expression of CD49b, CD51, CD29, and CD51/CD61 (**A**–**D**) in HCC-1954 and (**E**–**H**) in MCF-7 cell lines. Dotted grey lines are unstained controls. (**I**) shows the median fluorescence intensity (MFI) of three independent experiments. (**J**) Immunofluorescence of E-cadherin in HCC-1954 and MCF-7 cell lines. (**K**) Western blot of E-cadherin in total extract (TE), flow-through (FT) and membrane-enriched (ME) fractions of HCC-1954 and MCF-7 cell lines. (**L**) Western blot quantification of three independent experiments. *p*-value < 0.05 (*); <0.01 (**); <0.001 (***). Raw data of Western blot membrane are shown in Appendix A.

### 2.8. HCC-1954 Cell Line Presents Higher β1 Integrin Plasma Membrane Expression

As seen in Appendix A, the β-1 integrin band presented a higher intensity level in the ME fraction from HCC-1954 compared to the ME fraction of the MCF-7 cell line. Furthermore, this same band in the ME fractions of both cell lines was more intense than the corresponding band in the TE and FT fractions. This result demonstrated the efficiency of the membrane protein enrichment method used in our investigation.

### 2.9. β1 Integrin Presents a Significatively Higher Signal in the HCC-1954 Cell Line after Trastuzumab Treatment

To compare the differences in β1 integrin expression of cells treated or not with trastuzumab, we analysed the HER-2-overexpressing HCC-1954 that was resistant to trastuzumab treatment and the MCF-7 cell line. We also decided to include another HER-2-overexpressing cell line (BT-474) that was responsive to trastuzumab treatment. We first confirmed that only BT-474 was sensible to trastuzumab by measuring cell viability (MTT), as shown in Appendix A. The results (Figure 5, panel B) showed that the HCC-1954 treated with trastuzumab presented a significatively higher signal intensity of β1 integrin than the HCC-1954 not-treated cell. For the MCF-7 and BT-474 cell lines, two bands most likely corresponding to isoforms of β1 integrin were detected (Figure 5, panel A). However, no significative differences in the expression level of β1 integrin isoforms were observed between the treated and not-treated cell lines (Figure 5, panel B).

### 2.10. Trastuzumab Treatment Enhanced HCC-1954 Cell Migration

Since β1 integrin is related to cell migration we decided to investigate the effect of trastuzumab treatment on this function. For that we performed a wound-healing assay using the HER-2+ cell lines that were resistant and sensitive to the treatment, HCC-1954 and BT-474, respectively. The assay showed a statistically significant difference between both the control (not-treated) and treated cells. After 24 h HCC-1954 treated cells showed an enhanced migration, as evidenced by a smaller gap (%) compared to their control (not-treated) counterparts. On the other hand, BT-474 treated cells showed a bigger gap (%) compared to not-treated BT-474 cells (Figure 6). These results suggested that trastuzumab treatment enhanced HCC-1954 cell migration.

### 2.11. HER-2 and β1 Integrin Expression Presented a Trend in Correlation in HER-2+ but No Correlation in Luminal A Breast Cancer Subtypes

Transcriptome data deposited in TCGA were used to observe a possible correlation between the mRNA expression levels of HER-2 and β1 integrin in breast cancer samples. A weak correlation was observed between HER-2 and β1 integrin in HER-2+ breast cancer samples (Spearman’s r = 0.214, *p* = 0.0617, Table 2). In luminal A breast cancer samples, a negligible correlation between HER-2 and β1 integrin expression was found (Spearman’s r = −0.035, *p* = 0.4199, Table 2).

## 3. Discussion

Breast tumours differ in their biological characteristics and therefore in their ability to expand via metastasis [22,23,27]. Understanding these characteristics is essential for our ability to interfere in the evolution of the disease [6,24].

Here we compared in depth the proteomics profiles linked to the plasma membranes of two phenotypically distinct breast cancer cell lines, the highly invasive HCC-1954 and the non-invasive MCF-7. HCC-1954 is a poorly differentiated breast cancer cell line isolated from a grade 3 IIA primary invasive ductal carcinoma, with no lymph node metastases, that overexpresses HER-2. This membrane receptor participates in several biological processes such as cell growth, proliferation, survival, and invasion [25,26,28]. In addition, this cell is negative for ER/PR receptors and is resistant to trastuzumab treatment [18]. Most of the available target therapies are directed towards cell membrane proteins, as these proteins are more accessible and directly involved in metastasis [29,30,31].

The comparative proteomic approach highlights differences in the membrane protein expression profile that may affect their different mechanobiological capabilities. The proteomic analysis identified HER-2, β1 integrin, and E-cadherin with a higher expression in HCC-1954 than in MCF-7 ME fractions. The STRING interaction network (Figure 3) highlight a strong interaction between these proteins. Proteomaps and KEGG analyses (Figure 3) showed that proteins related to biological processes such as cytoskeletal organisation and tight junction proteins were more abundant in HCC-1954, whereas proteins related to proteasomes and histones were more abundant in MCF-7. These differences in cell protein machinery may be related to the greater capacity for migration in HCC-1954 and the higher rate of DNA mutation in MCF-7. Hampton et al., (2011) [32], showed that HCC-1954 and MCF-7 present amplified chromosomal segments, breakpoint clusters, and affected genes located at different positions across their genomes that correspond to different oncogenes, tumoral suppressor genes, and defects in DNA repair machinery. The authors suggested that the difference in structural mutability profiles between HCC-1954 and MCF-7 correlates with their differences in phenotype. A greater number of break- point DNA repair mutations were found in mutations of the MCF-7 and HCC-1954 cell lines associated with both HER-2 and cyclin D1 amplifications. Cyclin D1 amplification was associated with a high migration capacity and cell-cycle regulation at the G1 phase [32]. These results are in accordance with HCC-1954 and MCF-7 characteristics, proving that they represent the types of tumours from which they originated, an invasive ductal carcinoma and a poorly invasive one, respectively.

Indeed, the top pathways analyses of increased proteins in HCC-1954 were linked to motility: tight junctions, actin cytoskeleton regulation, and focal adhesion, processes that promote cancer invasion and metastasis. Metastasis accounts for over 90% of deaths in breast cancer, and motility is the first step for cell invasion to produce metastasis [31]. In fact, the HER-2-positive breast cancer subtype has an elevated risk for developing metastatic disease, and the trastuzumab treatment has improved the efficacy of chemotherapy and delayed the emergence of metastatic disease [33,34,35]. However, some patients are resistant or develop resistance within five years of trastuzumab treatment. It seems that integrin β1-associated signalling pathways may contribute to the progression of tumours that overexpress HER-2 [29,36]. This mechanism has yet to be proven. Nevertheless, our protein interaction network data reveal a strong interaction between HER-2 and both integrins αv and β1 with E-cadherin in the HCC-1954 cell line (Figure 3F).

We found a higher expression of both integrins αv and β1 in the plasma membrane of HCC-1954. Moreover, trastuzumab treatment enhanced integrin β1 expression and induced a faster migration rate for the HCC-1954 cell line, which was resistant to the treatment. This effect may justify the increased aggressiveness in treating trastuzumab-resistant patients with metastatic tumours [11,12]. The interaction between integrins αv, β1, and HER-2 has relevant clinical implications, since overexpression of integrin β1 has been identified as an independent marker of poor prognosis in patients with HER-2-positive breast cancer treated with trastuzumab [37,38]. Huang et al., (2011) [34] studied the role of β1 integrin on the mechanism of lapatinib (a protein kinase inhibitor) resistance in breast cancer and showed that proliferation of both HCC-1954 wild type and HCC-1954TL (resistant to lapatinib and trastuzumab treatment) were inhibited by AIIB2 (a monoclonal antibody that blocked β1 integrin).

Our analysis of the HER-2+ breast cancer molecular subtype in TCGA database showed that there is a trend of correlation between β1 integrin and HER-2 gene expression. On the other hand, this phenomenon was not observed in the luminal A molecular subtype. In vitro experiments showed that the HCC-1954 cell line (resistant to trastuzumab treatment) presented a higher level of β1 integrin expression after treatment with trastuzumab. On the other hand, in the MCF-7 cell line (no HER-2 overexpressed) and BT-474 cell line (sensitive to trastuzumab treatment) there were no significative differences in β1 integrin expression after trastuzumab treatment (Figure 5). Then we concluded that the HCC-1954 cell line could be used to study the involvement of β1 integrin on the mechanism of trastuzumab resistance.

The balance or interplay between integrins, extracellular matrix, and E-cadherin adherens junctions affects the mechanism by which invasive cells migrate [39,40,41] and the subsequent growth at a metastatic site [10,11,41]. Our proteomic data analysis showed that HCC-1954 had a high membrane E-cadherin abundance, which was validated by immunofluorescence (Figure 4J). Using a quantitative scoring system for the epithelial–mesenchymal spectrum classification, Le et al., (2018) [42] ranked the HCC-1954 cell line as having an intermediate epithelial–mesenchymal phenotype [42,43]. In fact, loss of E-cadherin expression or its cell membrane localisation was associated with a more aggressive and metastatic tumour, which acquired a mesenchymal phenotype. Although these processes suggest that loss of E-cadherin is implicated in the acquisition of a more invasive and metastatic phenotype, the loss of E-cadherin is not always a requirement for tumour progression [44,45]. Recent data suggest that this phenomenon is epithelial–mesenchymal plasticity (EMP) and that the most invasive cells are those that are in an intermediate stage between the epithelial and mesenchymal phenotypes [7]. In addition, invasion and migration can occur simultaneously, which seems to involve the expression of cadherins [13]. Therefore, an invasive tumour can display plasticity, enabling it to migrate as a single cell or as a cell group. This can be affected by the surrounding interactions or signals from the tumour environment [46]. Although MCF-7 is an epithelial cell, we found a more intense signal and abundance of E-cadherin in the FT fraction than in the ME fraction. Our result agreed with studies previously reported by Matsui et al., (2018) [47]. They showed that MCF-7 grown in a high-glucose culture medium enhances malignant characteristics such as anchorage-independent colony growth and Akt activation in MCF-7 cells. Both characteristics were followed by loss of contact inhibition and cells proliferating in multi-layers. In fact, we observed that the cell staining (DAPI) by confocal microscopy (Figure 4J) showed different nuclear intensities and sizes. This could indicate that cells were growing as multilayers without adherence to the glass coverslip. It is well established that E-cadherin is one of the major regulators of contact inhibition in cell proliferation [45]. Thus, the use of long-term high-glucose culture medium in our study could affect the amount of E-cadherin found in the MCF-7 membrane fraction.

We also identified CD166 as a protein found only in the HCC-1954 ME fraction. CD166 (also known as activated leukocyte cell adhesion molecule, or ALCAM) is a transmembrane immunoglobulin that has been reported to take part in resistance to therapies in cancer, contributing to tumour propagation and invasiveness [48]. ALCAM expression seems to have two types of patterns in breast cancer: patients with high cytoplasmic expression develop a more invasive tumour and suffer a shortened disease-free survival [49]. Extensive ALCAM expression in the plasma membrane triggers attenuated adherent ability which reinforces the motility of breast cancer cells, leading to metastasis [48]. Thus, an abundance of CD166 in the HCC-1954 ME fraction also contributes to the highly migratory phenotype of this cell line.

In summary, our results indicated that HCC-1954 and MCF-7 are quite different from the point of view of protein profile and abundance, and that these features reflect the differences in the individual phenotypes. We demonstrated that HCC-1954 expresses more HER-2, αv and β1 integrin, E-cadherin, CD166, and the CD44 stem cell marker. All these proteins appear to interact with each other to modify the cytoskeleton organisation and function, contributing to cell motility, invasiveness, and resistance to drug treatment. Moreover, the higher motility and membrane β1 integrin expression after trastuzumab treatment in the HCC-1954 cell line emphasises the need for investigating the contribution of β1 integrin modulation and its effect on the mechanism of trastuzumab resistance.

## 4. Materials and Methods

### 4.1. Cell Culture

Breast cancer cell lines MCF-7 (HTB-22) and HCC-1954 (CRL-2338) obtained from ATCC (American Type Culture Collection) were grown in RPMI-1640 medium (Sigma, St. Louis, MO, USA) supplemented with 10% FBS, 1 mM sodium pyruvate, 4.5 g/L glucose and antibiotics (sodium G penicillin 100 U/mL and streptomycin 100 µg/mL, Sigma-Aldrich, St. Louis, MO, USA) at 37 °C and 5% CO_2_. The BT-474 cell line (HTB20) was a gift from Professor Lídia Moreira Lima (Laboratório de Avaliação e Síntese de Substâncias Bioativas) from the Federal University of Rio de Janeiro–LASSBio/UFRJ); the cell was grown in DMEM high-glucose medium supplemented with 10% FBS, 1 mM sodium pyruvate, 0.01 mg/mL of insulin and antibiotics. Cells were routinely maintained until they reached confluence and then enzymatically detached with 0.125% Trypsin–0.78 mM EDTA (Sigma-Aldrich, St. Louis, MO, USA). The characteristics of each cell line are described in Appendix A. Appendix A shows a schematic diagram of the workflow used in this study.

### 4.2. Enrichment of Membrane Proteins, Quantification, and Protein Digestion

For membrane protein enrichment the Cell Surface Protein Isolation Kit (Pierce^®^, Waltham, MA, USA, cat. number 89881) was used exactly as described by the manufacturer. This kit enabled reducing the complexity of cellular total protein extracts; the membrane fraction was enriched with membrane proteins but also had other non-membrane proteins. In brief, four 75 cm^2^ flasks of confluent cells (90–95%) were washed 3 times with PBS (phosphate buffered saline) and incubated with Sul-fo-NHS-SS-Biotin for 30 min at 4 °C on a rocking platform. Then cells were scrapped into solution and transferred to a 50 mL conical tube, centrifuged at 500× *g* for 3 min, and the supernatant was discarded. After 3 washes with TBS (tris buffered saline), 500 µL of lysis buffer with protease inhibitor cocktail was added and cells were transferred to a micro-centrifuge tube. Cells were disrupted by sonicating with five pulses of 1 s burst at 1.5 watts, incubating 30 min on ice, and vortexing every 5 min for 5 s. After centrifugation at 10,000× *g* for 2 min at 4 °C the supernatant was collected and loaded onto the Neu-trAvidin Agarose column. The column was capped and incubated for 60 min at room temperature in a rotator platform. Then the column was centrifuged for 1 min at 1000× *g* and the flow-through fraction was collected. After 4 washes with washing buffer with protease inhibitor the membrane-enriched fraction was eluted with DTT 50 mM. This protocol generated two fractions from each cell line: the flow-through (FT) and the mem-brane-enriched (ME) fractions. Proteins were quantified by 2D-Quant Kit (GE Healthcare^®^, Chicago, IL, USA). Enzymatic hydrolysis of 100 µg of proteins was performed according to the FASP protocol as described by Wiśniewski et al., (2009) [50].

### 4.3. Peptide Desalting

The supernatants were desalted using the C18 reversed-phase column POROS R2 (Applied Biosystems, Waltham, MA, USA). The homemade columns were packed in a 200 µL tip containing a small piece of fiberglass. Resuspended C18 resin (100 µL at 40 mg/mL in isopropanol) was added to bring the column length to a few millimetres. The column was washed with 200 µL isopropanol, activated with 200 µL methanol, and conditioned two times with 200 µL of 0.1% TFA. The sample was then passed 5 times through the column. After 2 washes with 200 µL 0.1% TFA, the sample was eluted with 50% acetonitrile solution in 0.1% TFA. Samples were dried in a SpeedVac and resuspended at a concentration of 2 µg/µL in 5 mM ammonium formate buffer containing 5% acetonitrile pH 3.2.

### 4.4. Label-Free LC-MS^E^ Analysis

Each sample was analysed in quadruplicate on a two-dimensional liquid chromatography system (2D-SCX/RP-nanoAQUITY UPLC chromatography system, Waters, Farmington, MI, USA) coupled to a SYNAPT G1 HDMS™ mass spectrometer (Waters, Farmington, MI, USA). The samples were resuspended in an anion exchange loading buffer (5 mM ammonium formate and 5% acetonitrile, pH 3.2) to a final concentration of 1 µg/µL. A strong cation exchange column (SCX) (nanoACQUITY UPLC SCX TRAP 180 µm × 20 mm column packed with symmetrical particles of 5 µm diameter) (Waters, Milford, MA, USA) was used in the first dimension. For the second dimension, a nanoACQUITY BEH130 C18 reversed-phase analytical column (1.7 µm particle size, 130 Å pore size, and 75 µm × 150 mm dimension) (Waters, Milford, MA, USA) was employed. Approximately 2 µg of peptide mixture was injected, and 9 µL of a step gradient containing 8 fractions (“salt plugs”) of ammonium formate solution (50 to 350 mM) and acetonitrile (5 to 50%) was applied. The eluted peptides were then separated on the C18 reverse-phase analytical column with a linear gradient of 0.1% formic acid in water (mobile phase A) and 0.1% formic acid in acetonitrile (mobile phase B) for 80 min with a flow rate of 300 nL/min. The reversed-phase column was equilibrated with 95% A and 5% B, and the gradient was formed by a linear increase in concentrations of 5% to 50% B over 56.8 min, then an increase to 85% B after 3 min, and kept at 85% B for 3 min. The gradient was then decreased to 5% B within 3 min. Mass spectrum data were acquired in ion positive mode. The flight time analyser (TOF) was calibrated with [Glu1] fibrinopeptide B (GFP) ion fragments (Sigma-Aldrich, St. Louis, MO, USA) at a concentration of 100 fmol/µL at 50:50:1, methanol:H_2_O:acetic acid, in the range 50 to 2000 m/z. The GFP double charge precursor m/z ratio 785.8426 was used for lock mass correction. The MS^E^ (Waters, Milford, MA, USA) is a Data Independent Acquisition (DIA) methodology where the collision energy is alternated such that two channels are collected. The first channel includes the abundance measurements of the intact peptides, and a second channel is for the fragmented peptides; thus, every precursor is fragmented. Both are acquired at a high sampling rate. The collision energies alternated between high and low, remaining for 0.8 s (per scan) in each mode and 0.02 s in the scan interval. In low energy MS mode, the collision energy was 6 eV, and in high energy MS mode the collision energy increased from 15 to 55 eV. Fragmentation of the ions was performed by the collision-induced dissociation (CID) process that generates the MS/MS mass spectrum using argon as an inert gas for collision. The cone voltage of the ionisation source was 30 V.

### 4.5. Mass Spectra Data Analysis and Protein Quantification

Proteins were identified and quantified using Progenesis QI software v. 2.0 (Waters, Milford, MA, USA; program description is available at www.nonlinear.com, accessed on 3 August 2022). Mass spectrometry data files for each one of the four fractions, HCC-1954 ME (membrane-enriched), HCC-1954 FT (flow-through), MCF-7 ME, and MCF-7 FT, were obtained in nine raw archives from the corresponding gradient elution steps applied onto the anion exchange chromatography column. Proteomic analysis was evaluated with ANOVA <0.05 for each step separately, and then all nine steps analysed individually were combined using the application combine samples embedded in Progenesis QI. Mass spectrum analyses were performed using the revised and reverse human database from the UNIPROT database (http://www.uniprot.org/, accessed on 31 July 2019). For protein search carbamidomethylation of cysteines was considered as a fixed modification and the variable modifications considered were methionine oxidation, N-terminal acetylation, asparagine and glutamine deamidation, N-terminal carbamylation, N-terminal and C-terminal methylation. One trypsin miscleavage was considered acceptable. The precursor ion mass and fragment error tolerance were 10 and 20 ppm, respectively. Progenesis QI enabled a manual curation to guarantee the quality of peptide identification and protein quantification with peptide score selection, just one unique peptide and non-conflicting peptides for quantification. Our protein identification criteria were based on at least 2 ions per peptide fragment, 5 peptide fragments per protein, and 1 peptide per protein (non-conflicting peptides). The acceptable false discovery rate (FDR) was ≤4%, and a hit score >4.5 was used for filtering peptide sequence [51]. After the automatic chromatogram alignment, the precursor signal intensity was used for quantifications. Normalisation was performed using the method for “all proteins”; in this method the program automatically selected one run as the reference to normalise all proteins. The relative quantification was based on non-conflicting peptides with protein grouping; identified and quantified proteins were filtered with a 95% confidence (ANOVA *p* < 0.05). Proteins identified as unnamed as well as the reversed sequences were removed from the list of identified proteins.

### 4.6. Bioinformatic Analysis

A Venn diagram was constructed with InteractiVenn (http://www.interactivenn.net/, accessed on 17 May 2022). Gene ontology and pathway analysis were performed with Panther GO (pantherdb.org), KEGG Mapper (https://www.genome.jp/kegg/tool/map_pathway2.html, accessed on 23 January 2022) and Proteomaps (https://www.proteomaps.net/, accessed on 20 January 2022). Protein interactions were analysed with String (https://string-db.org/, accessed on 7 January 2021). Volcano plot and heat map analyses were carried out with Perseus v.1.6.10.50 and Morpheus (https://software.broadinstitute.org/morpheus/, accessed on 20 August 2021), respectively.

### 4.7. Immunofluorescence

HCC-1954 and MCF-7 (2 × 10^4^ cells) grown on glass coverslips were washed 3 times with PBS and fixed with 4% paraformaldehyde for 15 min at room temperature (RT). Fixed cells were washed again with PBS and blocked with 5% BSA (bovine serum albumin, Sigma) in PBS for one hour in a humid chamber at RT. Cells were incubated overnight in a humid chamber at 4 °C with the primary antibodies: polyclonal rabbit anti-ErbB2 (ab2428, Abcam, Cambridge, UK) or monoclonal mouse anti-E-cadherin (13-1700, clone HECD-1, Zymed laboratories). The antibodies were respectively diluted at 1:100 and 1:200 in PBS with 1% BSA. After two washes with PBS the secondary antibodies (Alexa Fluor 594 donkey anti-rabbit A21207, Invitrogen, or Alexa Fluor 488 goat anti-mouse A11001, Invitrogen, diluted at 1:200) were added and incubated in a humid chamber for one hour at RT. Immunofluorescence IgG controls were performed using only secondary antibodies. The cells were rinsed twice for 5 min each with PBS and twice with distilled water. Then the coverslips were mounted on microscope slides with ProLong Gold antifade reagent with DAPI (P36931, Invitrogen, Waltham, MA, USA). High resolution images were captured at 20× magnification using a Zeiss microscope (Carl Zeiss Microscopy, Munich, Germany).

### 4.8. Flow Cytometry Analysis (FACS)

Cells were submitted to a mild trypsinisation with a solution of 0.125% trypsin and 0.78 mM EDTA (Sigma-Aldrich, St. Louis, MO, USA). Cell detachment from the culture flask was monitored under an inverted microscope, and the enzymatic reaction was immediately inhibited by adding culture medium supplemented with 10% FBS. The cells were centrifuged at 1200 rpm for 5 min at 4 °C, the pellet was resuspended in culture medium supplemented with 10% FBS, and cells were quantified in a hemocytometer with trypan blue to evaluate viable cells. For FACS, approximately 5 × 10^5^ cells were washed twice with cold PBS containing 3% FBS and 0.1% sodium azide (FACS buffer) and incubated for 30 min on ice with the following directly conjugated antibodies: mouse anti-human CD29-Alexa 700 (clone TS2/16), CD51 R-PE (R-phycoerythrin, clone NKI-M9), CD49b-FITC (fluorescein, clone P1E6-C5), CD51/CD61-FITC (clone 23C6) from Biolegend and CD24-PE (clone SN3, Thermo Fischer, Waltham, MA, USA) and CD44-APC (clone IM7, eBioscience, San Diego, CA, USA). The cells were washed twice with FACS buffer. Acquisition was performed on a FACSCanto-II (BD Biosciences, Franklin Lakes, NJ, USA) using the software Diva 6.1.3 (BD Biosciences, Franklin Lakes, NJ, USA). Data were analysed using Diva and FlowJo (Tree Star Inc., Ashland OR, USA).

### 4.9. Western Blotting of HER-2 and E-Cadherin in Fractions Obtained from the Cell Surface Protein Isolation Kit from Pierce^®^

All fractions obtained from HCC-1954 and MCF-7 cell lines with the Cell Surface Protein Isolation Kit from Pierce^®^ (total protein extract, FT, and ME) were assayed by Western blot. A total of 20 µg of each fraction was loaded on each lane of an 8% SDS-PAGE gel. Following electrophoresis, the protein bands were transferred overnight to a PVDF membrane, and the blot was probed with the following antibodies: rabbit anti-human ErbB-2 (ab-2428, Abcam) diluted to 1:1000, and mouse anti-human E-cadherin (610182, BD) diluted to 1:20,000. The β-actin antibody (A2228, Sigma) diluted to 1:5000 was used to normalise gel loading. IRDye 800CW (cat. number 92532213, Li-COR) was used as a secondary antibody. Membranes were imaged and quantified with a Li-COR Odyssey Infrared Imaging system (Li-COR Bioscience, Cambridge, UK).

### 4.10. Western Blotting of β1 Integrin from Cell Lines Treated or Not with Trastuzumab and from Cell Line Fractions

The cell lines HCC-1954, MCF-7, and BT-474 with 80% of confluence were incubated for 24 h without FBS, then cells were treated with 20 µg/mL of trastuzumab (Herceptin^®^) for 72 h [27]. Controls were performed with untreated cell lines. Thereafter, total proteins were extracted with RIPA buffer (50 mM Tris HCl, 150 mM NaCl, 1.0% (*v*/*v*) NP-40, 0.5% (*w*/*v*) Sodium Deoxycholate, 20 mM NAF, 0.2% (*w*/*v*) SDS, 1 mM orthovanadate and protease inhibitor cocktail) and quantified with the BCA Protein Quantification Kit (Pierce). The total proteins extracted from treated and untreated cell lines and from TE, FT, and ME fractions were assayed by Western blot for β1 integrin expression. In brief, 30 µg of each extract was loaded on an 10% SDS-PAGE gel. Following electrophoresis, the protein bands were transferred overnight to a PVDF membrane, and the blot was probed with the mouse anti-human β1 integrin (ab52971, Abcam) diluted to 1:10,000. The anti-actin antibody (clone C4 MAB 1501, Millipore) diluted to 1:1000 was used to normalise gel loading. The anti-rabbit IgG (A0169, Sigma) and the anti-mouse IgG (A2304) peroxidase conjugated diluted 1:20,000 were used as secondary antibodies. The chemiluminescent reagent (Pierce™ ECL Western Blotting Substrate, cat. number 32209) was used to revelled bands. Membranes were imaged with ChemiDoc imaging system (BioRad Laboratories, Inc., Hercules, CA, USA) and quantified using ImageJ.

### 4.11. Wound-Healing Assay

HCC-1954 or BT-474 cells at a density of 2.5 × 10^4^ cells/well were plated in 24-well plate in five replicates. After cells reached 90% confluence, trastuzumab at a concentration of 20 µg/mL in complete medium was added to the treated group, the control group received only fresh medium, and cells were incubated for 72 h at 37 °C in a CO_2_ incubator. After this, the cell monolayers were scratched with a 200 μL pipette tip. The cells were washed with PBS to discard cell debris and then replaced with fresh medium. Images of the point time zero were acquired immediately and after 24 h of incubation using an Olympus CKX41 microscope coupled with an EP50 camera. Five images were acquired for each condition at each time point. The wound area was measured using Image J (Fiji software). Data are presented as percentages of the remaining gap after 24 h calculated using the following formula: Gap % = (gap area at 24 h × 100)/gap area at 0 h.

### 4.12. Gene Expression Correlation Analysis

Transcriptome data from 535 luminal A breast cancer samples and 77 HER-2-positive breast cancer samples were collected from The Cancer Genome Atlas (TCGA), Firehose Legacy—previously known as TCGA Provisional; http://cancergenome.nih.gov/, accessed on 4 October 2021. In the current study, we analysed the correlation between the expression of integrin beta-1 (*ITGB1* gene) and the expression of HER-2 (*ERBB2* gene) in both breast cancer subtypes. The mRNA expression levels in FPKM (frames per kilobase per million) units were downloaded.

### 4.13. Statistical Analysis

Results were analysed using GraphPad Prism 8 (GraphPad Software, San Diego, CA, USA, accessed on 3 August 2022) and expressed as mean ± SEM. The results represent the average of three independent experiments (n = 3). One-way analysis of variance (ANOVA) followed by Tukey’s post-test was used for Western blot analysis. The unpaired Student’s *t*-test was applied for FACS assays. RNA-seq values from TCGA database were correlated and statistically analysed with the non-parametric Spearman test. *p*-values less than 0.05 were considered statistically significant.

## Figures and Tables

**Figure 1 ijms-23-10194-f001:**
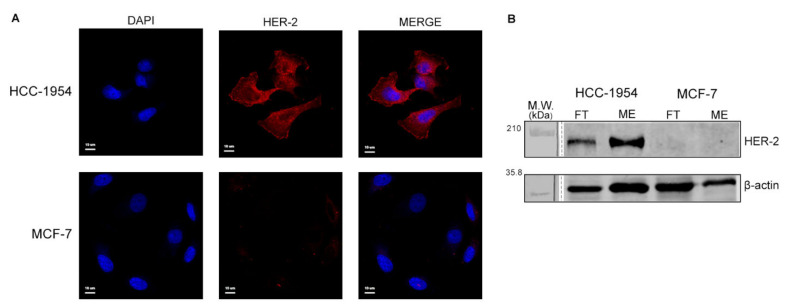
HER-2 protein expression analysis in HCC-1954 and MCF-7 cell lines. (**A**) Immunofluorescence of HER-2 in HCC-1954 and in MCF-7 cell lines. (**B**) Western blot of HER-2 in total extract (TE), flow-through (FT) and membrane-enriched (ME) fractions of HCC-1954 and MCF-7 cell lines. Raw data of Western blot membrane are shown in Appendix A.

**Figure 2 ijms-23-10194-f002:**
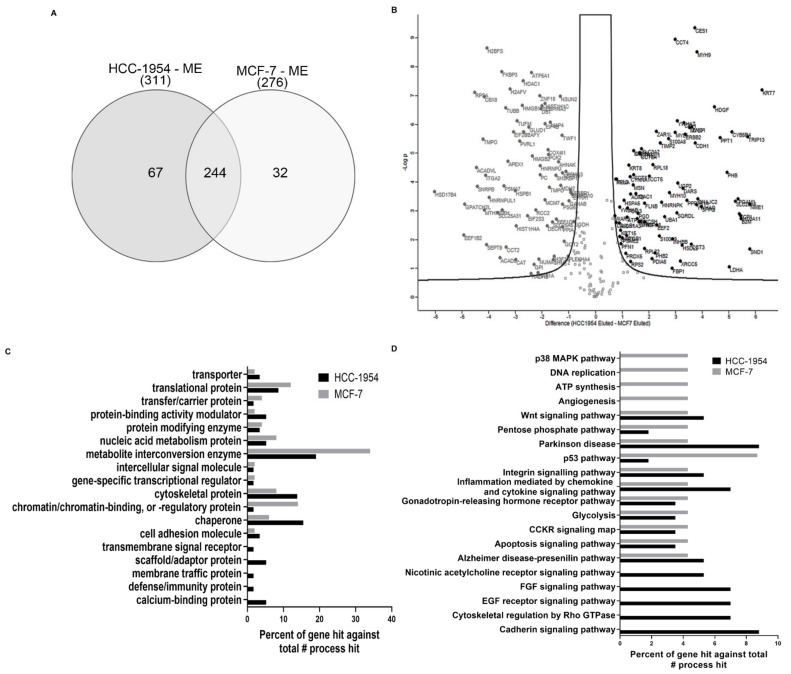
Proteomic data Analysis (**A**) Interactive Venn diagram of the 343 proteins identified and quantified with a 95% confidence level (ANOVA *p* ≤ 0.05) in the Progenesis QI analysis of our label-free proteomic data of HCC-1954 and MCF-7 ME fractions. (**B**) Volcano plots of all proteins from HCC-1954 and MCF-7 ME fractions. Proteins with increased fold change ratio are indicated by black circles, whereas the grey circles denote those proteins presenting a decreased fold change ratio (Data are available in Appendix A). (**C**) GO Panther classification of increased proteins with a relative fold change higher than 2 (0.5 < FC > 2) for the ratio HCC-1954 ME/MCF-7 ME in the Protein class and (**D**) Pathway Classification.

**Figure 3 ijms-23-10194-f003:**
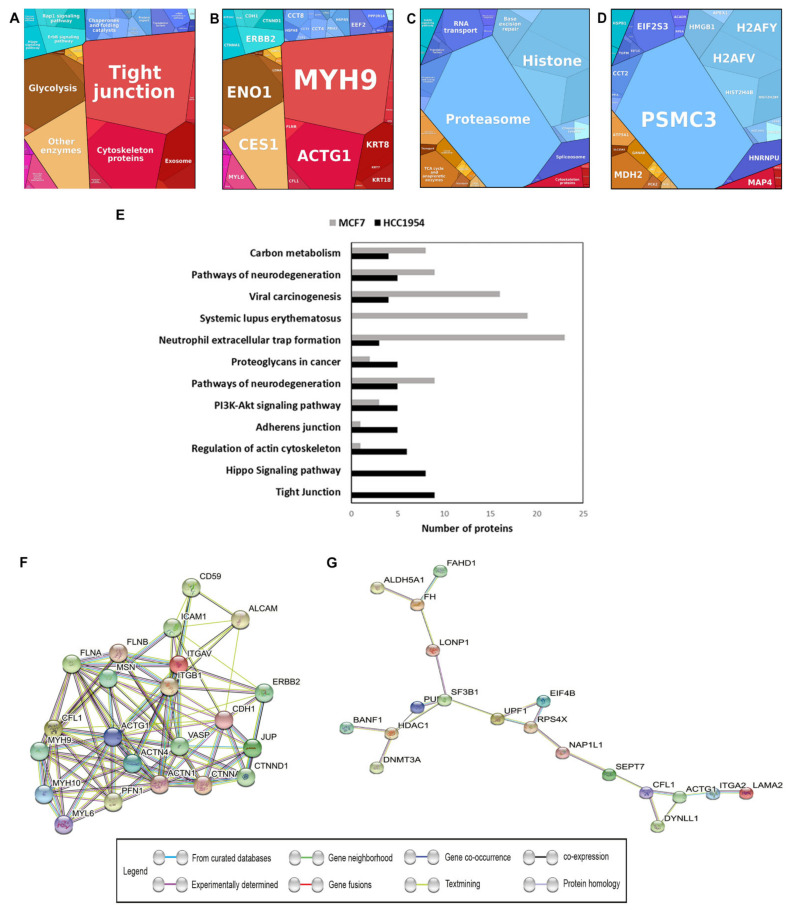
Proteomaps and top pathway analysis of increased proteins with a relative fold change higher than 2 (0.5 < FC > 2) for the ratio HCC-1954 ME/MCF-7 ME. (**A**,**C**) show the biological processes analysed for HCC-1954 and MCF-7, respectively. (**B**,**D**) show the proteins classified in each case. (**E**) Top pathways of ME fraction protein classification from HCC-1954 and MCF-7 performed with the Kegg Mapper platform. Interaction networks of proteins increased (**F**) in HCC-1954 ME and (**G**) in MCF-7 ME. HCC-1954 with a PPI enrichment *p*-value of < 1.0 × 10^−16^ and MCF-7 with a PPI enrichment *p*-value of 4.48 × 10^−5^, respectively.

**Figure 5 ijms-23-10194-f005:**
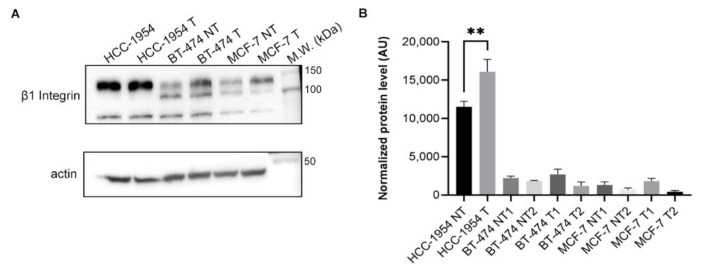
β1 Integrin expression in the HCC-1954, BT-474, and MCF-7 cell lines treated (T) or not treated (NT) with trastuzumab. (**A**) Western blot of β1 integrin showing different expression levels in HCC-1954, BT-474, and MCF-7 cell lines in not-treated (NT) and treated (T) with 20 µg/mL of trastuzumab for 72 h. (**B**) Western blot quantification of β1 integrin in three independent experiments. Bar chart shows quantification of protein levels compared to the control in each condition. Actin was used as the load control. Error bars show standard deviation, ** *p*  < 0.01. Raw data of Western blot membrane are shown in Appendix A.

**Figure 6 ijms-23-10194-f006:**
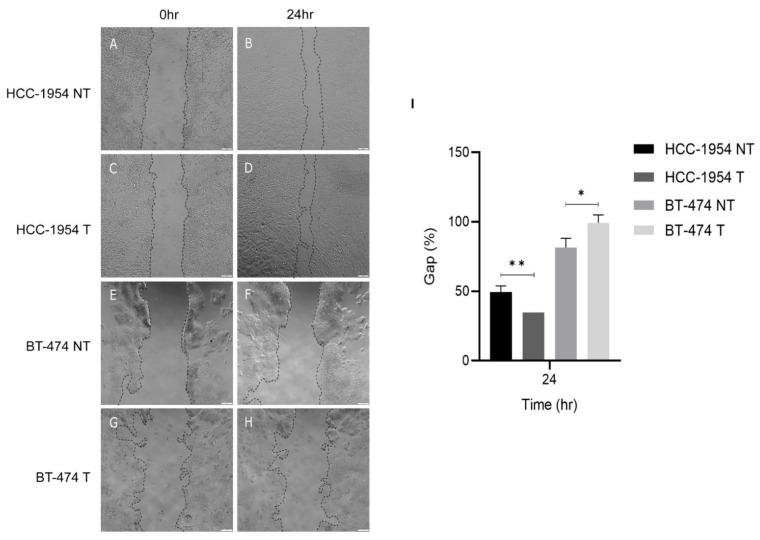
Trastuzumab effect on HCC-1954 and BT-474 migration potential. (**A**–**D**) Images of HCC-1954 cells and (**E**–**H**) BT-474 cells. (**A**,**C**,**E**,**G**) when the scratch/wound was performed, and (**B**,**D**,**F**,**H**) 24 h after. (**I**) Quantification of the gap (%). (NT) not-treated and (T) treated. These results show an enhanced migration of HCC-1954 cells with treatment, as evidenced by a smaller gap after 24 h compared to not-treated cells. On the other hand, treated BT-474 cells presented a larger gap after 24 h compared to not-treated cells. Scale bars correspond to 100 µm. *p*-value < 0.05 (*); <0.01 (**).

**Table 2 ijms-23-10194-t002:** Correlation Analysis of HER-2 and β1 Integrin in Human Breast Cancer Patients of Luminal A and HER-2+ Subtypes using Data from the Cancer Genome Atlas (TCGA).

	Gene 1	Gene 2	Spearman r	*p* Value
** *Luminal A* **	** *ITGB1* **	** *HER2* **	−0.035	0.4201
** *HER-2^+^* **	** *ITGB1* **	** *HER2* **	0.214	0.0617

Luminal A: N = 535 samples; HER-2+: N = 77 samples.

## Data Availability

The mass spectrometry proteomics data were deposited in the ProteomeXchange Consortium via the PRIDE [52] partner repository with the dataset identifier PXD023876, at [http://proteomecentral.proteomexchange.org/cgi/GetDataset] accessed on 15 January 2021.

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
