# Peer review of "Proteomic Analysis of HCC-1954 and MCF-7 Cell Lines Highlights Crosstalk between αv and β1 Integrins, E-Cadherin and HER-2"

_ijms, 2022, doi:10.3390/ijms231710194_

Round 1
Reviewer 1 Report (New Reviewer)
This manuscript describes a proteomic analysis in two different breast cancer cell lines, one HER2+ and one HER2-. The authors perform label-free proteomic analysis in which they highlighted that molecules which are related with cytoskeleton organization and are highly expressed in HER2+ cell line compare to HER2- cell line. Among these molecules is b1 integrin, which is overexpressed after treatment with trastuzumab and may be related to resistance to trastuzumab mechanism. The design and the experimental part scientifically sound reasonable and the manuscript is well-written. However, the scientific outcomes of this work focused on the cross-talk of b1 integrin with HER2 on HER2 resistant breast cancer. This finding was already known as the authors mentioned (Lesniak et al., 2009, Cancer Res, Toscani et al., 2017 Plos one) and also the cross-talk between integrins with growth factor receptors and syndecans is also known (see Michael Simons & Alan Rapraeger work). Moreover, the heterodimer of integrins as well as GFR can be differentiated on the different cells line sources. The authors have to perform more functional experiments in order to elucidate the role of cross-talk in HER2/b1 integrin in resistance to trastuzumab treatment and present which cell functions are affected from the treatment. In addition, the title should be restructured in order to be presented properly the findings of this work
Author Response
This manuscript describes a proteomic analysis in two different breast cancer cell lines, one HER2+ and one HER2-. The authors perform label-free proteomic analysis in which they highlighted that molecules which are related with cytoskeleton organization and are highly expressed in HER2+ cell line compare to HER2- cell line. Among these molecules is b1 integrin, which is overexpressed after treatment with trastuzumab and may be related to resistance to trastuzumab mechanism. The design and the experimental part scientifically sound reasonable and the manuscript is well-written. However, the scientific outcomes of this work focused on the cross-talk of b1 integrin with HER2 on HER2 resistant breast cancer. This finding was already known as the authors mentioned (Lesniak et al., 2009, Cancer Res, Toscani et al., 2017 Plos one) and also the cross-talk between integrins with growth factor receptors and syndecans is also known (see Michael Simons & Alan Rapraeger work). Moreover, the heterodimer of integrins as well as GFR can be differentiated on the different cells line sources. The authors have to perform more functional experiments in order to elucidate the role of cross-talk in HER2/b1 integrin in resistance to trastuzumab treatment and present which cell functions are affected from the treatment. In addition, the title should be restructured in order to be presented properly the findings of this work
We used the pdf archive of our manuscript where changes were highlighted in order to indicate the corresponding lines.
Re: We agree with your considerations and to show how the increased expression of Beta1 integrin would change the cell behavior we decided to carry on a wound healing assay. For that we used two HER-2+ breast cancer cell lines, HCC-1954 that is resistant and BT-474 that is sensitive to trastuzumab treatment. The results showed that after trastuzumab treatment HCC-1954 cell line presented a faster migration rate than without treatment, which could justify the increase of aggressiveness in trastuzumab resistance patients with metastatic tumors (Valabrega et al. 2007; Kunte et al, 2020 and Henriques et al, 2021). We concluded that the cell migration function was affected after HCC-1954 trastuzumab treatment and that HER-2 and beta 1 integrin may have a pivotal role in this process. These results were added to the manuscript (lines 483-492). The methodology is described at line 285-295. We also discussed the consequences of this effect at the discussion section lines 549-552. The follow up of this article will be to explore the mechanism involved in this crosstalk.
- -Valabrega G, Montemurro F, Aglietta M. Trastuzumab: Mechanism of action, resistance and future perspectives in HER2-overexpressing breast cancer. Vol. 18, Annals of Oncology. 2007. p. 977–84.
- Siddharth Kunte, Jame Abraham, Alberto J Montero. Novel HER2-targeted therapies for HER2-positive metastatic breast cancer. Cancer. 2020 Oct 1;126(19):4278-4288. doi:10.1002/cncr.33102.
- -Beatriz Henriques, Fernando Mendes and Diana Martins. Immunotherapy in Breast Cancer: When, How, and What Challenges? Biomedicines 2021, 9(11), 1687; https://doi.org/10.3390/biomedicines9111687.
Reviewer 2 Report (New Reviewer)
The paper entitled "Proteomic analysis of HCC-1954 and MCF-7 cell lines high-2 lights crosstalk between αv and β1 integrins, E-cadherin and 3 HER-2" the authors performed a label-free quantitative proteomic analysis of membrane proteins from HCC-1954 and MCF-7 cell lines, uncovering proteins and potential mechanistic explanation for invasive and metastatic behaviours in breast cancer. The topic and research approaches are sound, and may be of interest for a wider community of scientists. However that are several issues that must be corrected/ improved in other to increase the overall accuracy and quality of the paper, as follows:
- Clarify the use of "Label free LC-MSE analysis" rather than the most conventional Label-free LC-MS/MS analysis.
- Please clarify the use of 4% false discovery rate instead of the 1% FDR which is standard in label-free proteomics.
- Clarify the type of label-free quantification used in this paper (spectral counting or precursor signal intensity?)
- Indicate the normalisation methods performed in Progenesis QI.
- Justify the filtering based on:
Proteins identified as unnamed as well as the reversed ones were removed from the list of identified proteins.
- Clarify the meaning of "reserved ones".
- Based on what is described in the paper data dependant acquisition (DDA) was used for label-free proteomics analysis. It is well known that DDA generates many missing values do to the stochastic nature of precursor selection for MS/MS fragmentation. Please clarify the methods used to handle missing values.
- In figure 2a explain the 67 and 32 exclusive protein detected in HCC-1954 and MCF-7, respectively. In the reviewer's opinion this is a consequence of the missing values problem mentioned before.
- The total number of quantified proteins seems to be rather low, in comparison to what should be expected from LC-MS/MS analysis of cells.
- Indicate the total amount of protein digested using the FASP protocol.
- The gene ontology analysis shows general pathways of cell metabolism and homeostasis. It would be useful to the reader to have a perspective how the detected proteins contribute to what is generally described as cancer hallmarks.
Author Response
The paper entitled "Proteomic analysis of HCC-1954 and MCF-7 cell lines high-2 lights crosstalk between αv and β1 integrins, E-cadherin and 3 HER-2" the authors performed a label-free quantitative proteomic analysis of membrane proteins from HCC-1954 and MCF-7 cell lines, uncovering proteins and potential mechanistic explanation for invasive and metastatic behaviours in breast cancer. The topic and research approaches are sound, and may be of interest for a wider community of scientists. However that are several issues that must be corrected/ improved in other to increase the overall accuracy and quality of the paper, as follows:
Clarify the use of "Label free LC-MSE analysis" rather than the most conventional Label-free LC-MS/MS analysis.
We used the pdf archive of our manuscript where changes were highlighted in order to indicate the corresponding lines.
Re: In the most conventional label-free LC-MS/MS analysis (Orbitraps) the method used is Data Dependent Analysis (DDA). Here, we used another label-free methodology for quantitation of proteins by a Data Independent Analysis (DIA) method termed LC-MSE. MSE, developed by Waters, is a unique acquisition mode where the collision energy is alternated such that two channels are collected. The first channel includes the abundance measurements of the intact peptides and a second channel for the fragmented peptides thus, every precursor is fragmented. Both are acquired at a high sampling rate. This information was included on lines 173 to 177.
Please clarify the use of 4% false discovery rate instead of the 1% FDR which is standard in label-free proteomics.
Re: About the FDR, we totally understand their observations, normally these parameters (FDR<1) are used by programs like Maxquant, which is more suitable for data generated by the Thermo Orbitrap and FT mass spectrometers. However, in this work, we used the Progenesis QI Program to analyze the label-free mass spectrometry raw data generated by Synapt G1 spectrometer from Waters Co. The Progenesis QI (Nonlinear Dynamics a Waters company) compiles bioinformatic tools, that allow the use of pipelines with the FDR <4%, which were: a "resolving conflicts" curatorship step regarding the MS/MS where the user can do manual curation and guaranteed the quality of the identification and quantification with just one unique and non-conflicting peptide. It was also ensured in the software that the hits had a minimum score value (> 4.5), thus we have high trustworthiness in the results. Furthermore, we use ANOVA <0.05 for the selection of statistically significant quantified proteins. For clarity, we adjusted the method description on lines 204 to 208.
Clarify the type of label-free quantification used in this paper (spectral counting or precursor signal intensity?)
Re: The label-free quantification methodology used was based on non-conflicting peptides with protein grouping, the methodology of MSE analysis that is based on precursor signal intensity was used. These pieces of information were included on lines 199 - 202).
Indicate the normalisation methods performed in Progenesis QI.
The normalization method performed in Progenesis QI was “all proteins”. In this method, the program automatically picks out one run as the reference to normalize all proteins. It is an approach that is used to quantify the abundance of all proteins without relying on housekeeping proteins. This information was included on lines 204 – 208 (program description is available at www.nonlinear.com line 186)
Justify the filtering based on: Proteins identified as unnamed as well as the reversed ones were removed from the list of identified proteins. Clarify the meaning of "reserved ones
The SwissProt database used in our work is revised and non-redundant, so it contains a consensus sequence for each distinct protein, and the known variants have been collapsed into a single entry. In order to augment the assurance of the search we have added to the database the reversed sequences format of each distinct protein, this is a very simple and powerful way of validating search results. The search is done using identical search parameters, against a database in which the sequences have been reversed or shuffled. So, the reversed sequences found were deleted as it represents false protein sequences (we emphasize that the number of false proteins is below the FDR). The unnamed proteins are proteins that were translated from c-DNA sequences however nothing is known about their functions. The term “reversed ones” was changed to “reversed sequence” to be more precisely understood in line 211.
Based on what is described in the paper data dependant acquisition (DDA) was used for label-free proteomics analysis. It is well known that DDA generates many missing values do to the stochastic nature of precursor selection for MS/MS fragmentation. Please clarify the methods used to handle missing values.
Re: It is very important to clarify that we did not use a DDA method. The LC-MSE, used in this study, is a type of data-independent acquisition DIA and not a DDA. Therefore, all precursors are fragmented. In order to make it clearer in the manuscript, we added this information on Lines 173- 177.
In figure 2a explain the 67 and 32 “exclusive protein detected in HCC-1954 and MCF-7” respectively. In the “reviewer's” opinion this is a consequence of the missing values problem mentioned before.
Re: These proteins were identified and quantified only in HCC-1954 or MCF-7 ME fraction. It’s not a consequence of missing values since we used DIA as it we justified above.
The total number of quantified proteins seems to be rather low, in comparison to what should be expected from LC-MS/MS analysis of cells.
Re: The total number of quantified proteins was 1,386. After statistical analysis, 450 proteins were filtered to p<0.05 (ANOVA) using Progenesis QI. This information was included on page 7 on lines 348 to 351.
Indicate the total amount of protein digested using the FASP protocol.
Re: We thank the referee to call our attention to this missed information. The total amount of protein digested using FASP was 100 µg. This information was included in line 137.
The gene ontology analysis shows general pathways of cell metabolism and homeostasis. It would be useful to the reader to have a perspective on how the detected proteins contribute to what is generally described as cancer hallmarks.
Re: We take into account this observation and we included a paragraph and a reference in the results section on lines 404-409.
Reviewer 3 Report (New Reviewer)
Manuscript study positive and negative HER2 breast cancer cell lines and how difference in membrane protein production in those cell lines reflect on invasive pattern in breast cancer. From many analyzed proteins authors focus on receptor proteins - integrins and epithelial adhesion molecules like E cadherins. Researchers are using traditional experiments as western blot, Immunofluorescence, Mass spectra analysis of proteins, FACS analysis and modern approached as very useful software analyzing connections between proteins as proteo-maps and STRING which are very well and clearly presented.
Authors many times use invasion and migration to explain different pattern of membrane proteins in two cell lines and their effect on those phenotypical changes, although paper do not contain any functional data to connect them. In my opinion functional data would increase significant value of this paper and it would be worth to add migration assays with connections to E cadherin’s and integrin’s .
Overall I enjoyed reviewing this manuscript especially reading very well written dissection. I accept this paper to be publish after addressing my comments which are seen below.
Minor changes
- Please add molecular weight to your western blots as you have done it on the original western blots and in supplementary figures.
- Please provide IgG controls for your IF to prove specificity of the staining.
- Could you please provide viability assay for use of trastuzumab for MCF7 cells HCC1954, published references for use of drug for the same cell line is acceptable. Provided references show viability assays for different cell lines, most of the cell lines have different doses responses for the same drugs. Von Heyde S Der, Wagner S, Czerny A, Nietert M, Ludewig F, Salinas-Riester G, et al. mRNA profiling reveals 627 determinants of trastuzumab efficiency in HER2-positive breast cancer. PLoS ONE. 2015;10 (2):1–27.
- Could you please provide quantification analysis with original numbers and counting of densitometry data for fig5 as supplementary data in excel sheet. Shown western blot (Fig5A) does not reflect changes showing on the graph (Fig5B). It would be more convincing to see those big changes with real numbers. It has probably something to do with not equal beta actin in your western blots. Again I am not sure why beta actin is so not equal if you load the same concentrations. Maybe some problems with transfer?
- S fig 8 D- change please beta actin from 43kDa to 42kDa as all above. Why your beta actin in D has two bands? I do not see it in any other provided blots.
Major changes
- Fig1 A, IF for HER2, The fluorescent staining of HER2 should be more visual on the membrane. Staining is defused with the same intensity through all cell, what bring concern about its specificity. Provide please proper stain cells with visual membrane staining of HER-2.
- Fig1 B, Using beta actin as housekeeping gene for membrane protein is incorrect and show strong contamination of the sample with other no membrane molecules. I am aware that there are publication using beta actin for membrane and also nuclear fractionation but majority of scientists consider it incorrect. Please proof that your membrane fraction is clean and specific.
- Fig1 B Since the same concentration of protein 20ug was put in each section, proper housekeeping protein should be use to confirm proper loading. Shown western blot does not confirm it. Please use different housekeeping gene appearing equality in all lines. It is also acceptable to use transfer stain membrane as a proof of lading the same protein concentration.
- Fig1 B Can you please explain why your total extract for HER-2 detection has no HER-2 in HCC cell line?
- The same situation in Figure- S8 D looking at the beta integrin’s very strongly present in Membrane fraction but non seen in total extract.
- C and D. Can you explain why description of the figure is on protein analysis and the graphs marking saying genes? Genes level not always correspond with protein levels.
Author Response
Please see the attachment

Reviewer 4 Report (New Reviewer)
This manuscript is more likely a descriptive study and lacks validation of the connections among HER-2, αv and β1 integrins, E-cadherin, and breast cancer progression. Other major comments include:
- Proteomic comparison was conducted between HCC-1954 and MCF-7, two breast cancer cell line derived from different patients with different genetic characterizations which could have a large effect on the supposed results. Therefore, the initial comparison was not suitable, and the results seemed to be unreliable.
- Table 2, not clear how the authors divided the breast cancer patients (TCGA) into luminal A and HER-2+ Subtypes. The correlation between HER-2 and β1 integrin expression was not significant.
- How about the difference of E-cadherin, αv and β1 integrins expression between HER-2+ and HER-2- breast cancer patients? Their correlation with TNM stage and metastasis status?
- Will HER-2 regulate E-cadherin, αv and β1 integrins expression? If yes, what is the underlying mechanism? If not, how to explain their correlation?
Author Response
This manuscript is more likely a descriptive study and lacks validation of the connections among HER-2, αv and β1 integrins, E-cadherin, and breast cancer progression. Other major comments include:
- Proteomic comparison was conducted between HCC-1954 and MCF-7, two breast cancer cell line derived from different patients with different genetic characterizations which could have a large effect on the supposed results. Therefore, the initial comparison was not suitable, and the results seemed to be unreliable
We used the pdf archive of our manuscript where changes were highlighted in order to indicate the corresponding lines.
We respectfully disagree with the referee. The aim of this study was exactly to compare the breast cancer cell lines with different genetic and functional characterizations, to identify and quantify the proteins that would be related to the behavior differences between these cells. Furthermore, these are common practices used for proteomic and other studies of protein expression. Many articles use this approach for understanding cellular differences as we exemplified by these articles listed below that specifically uses the breast cancer cell lines MCF-7, T47D, and MDA-MB-231.
- Aka JA, Lin SX. Comparison of functional proteomic analyses of human breast cancer cell lines T47D and MCF7. PLoS One. 2012;7(2):e31532. doi: 10.1371/journal.pone.0031532. Epub 2012 Feb 24.
- Chareeporn Akekawatchai, Sittiruk Roytrakul, Narumon Phaonakrop, Janthima Jaresitthikunchai, Sarawut Jitrapakdee Proteomic Analysis of the Anoikis-Resistant Human Breast Cancer Cell Lines. Methods Mol Biol. 2020; 2138:185-193. doi: 10.1007/978-1-0716-0471-7_11.
- Table 2, not clear how the authors divided the breast cancer patients (TCGA) into luminal A and HER-2+ Subtypes. The correlation between HER-2 and β1 integrin expression was not significant.
We would like to elucidate that the classification of breast tumors into molecular subtypes was not performed by the authors. The authors downloaded the TCGA spreadsheet with transcriptomic data from these tumors. In this Excel spreadsheet, each patient already presented the molecular classification.
The correlation is not significant, but there is a trend in HER2+ breast tumors (P = 0.0617). Furthermore, we must mention that the number of patient samples with HER2+ breast tumors at TCGA (n = 77) is much smaller than that of patients with luminal A breast tumors (n = 535). In order to be more precise, we have changed the text accordingly. On lines 493-494 and line 561.
How about the difference of E-cadherin, αv and β1 integrins expression between HER-2+ and HER-2- breast cancer patients? Their correlation with TNM stage and metastasis status?
Re: The downloaded spreadsheet of TCGA with transcriptomic data from these tumors, does not include the TNM stage and metastasis information of the patient's tumors. Therefore, unfortunately, we could not make these correlations analyses.
Will HER-2 regulate E-cadherin, αv and β1 integrins expression? If yes, what is the underlying mechanism? If not, how to explain their correlation?
Re: Our protein-protein interaction network (figure 3F), an in silico analysis, showed a higher interaction between HER-2, E-cadherin, αv and β1 integrins (PPI enrichment p- value<1.0 e-16). This correlation corroborates the literature showing that HER-2 expression in mammary epithelial cells induces EMT and that process initiates prior to the downregulation of E-cadherin (Jenndah et al, 2005). The transformed mesenchymal cells display elevated levels of β1 integrin and low levels of HER-2 (Lesniak et al, 2013). Our data shows that HCC-1954 seems to have the same behavior.
Here we show that when the HER-2 + cells are treated with trastuzumab (a HER blocker) beta 1 integrin has a higher expression. Thus, we performed a wound healing assay with HCC-1954 cell line not treated and treated with trastuzumab. The results showed that the migration rate was higher in HCC-1954 after trastuzumab treatment. We believe that this response could be another mechanism of trastuzumab resistance. The follow-up of this article will be to explore the mechanism involved in this crosstalk and the involvement of E-cadherin.
- LE Jenndah, Petter Isakson, Dan Baeckström. -erbB2-induced epithelial-mesenchymal transition in mammary epithelial cells is suppressed by cell-cell contact and initiated prior to E-cadherin downregulation. Int J Oncol. 2005 Aug;27(2):439-48c
- D Lesniak, Siham Sabri, Yaoxian Xu, Kathryn Graham, Pravin Bhatnagar, Mavanur Suresh, Bassam Abdulkarim.Spontaneous Epithelial-mesenchymal transition and resistance to HER-2-targeted therapies in HER-2-positive luminal breast cancer. PLoS One. 2013 Aug 26;8(8): e71987. doi: 10.1371/journal.pone.0071987. eCollection 2013.
We have introduced this experiment in the manuscript. in the methodology on lines 285-295, in the results on lines 483-492, and changed discussion accordingly on lines 549-552 and line 615.
This manuscript is a resubmission of an earlier submission. The following is a list of the peer review reports and author responses from that submission.
Round 1
Reviewer 1 Report
In the presented manuscript, the authors described proteomic characterization and differentiation of HCC-1954 and MCF-7 cell lines. This topic is very interesting and up-to-date. The study is well designed, and the results are clearly presented and described in detail. My remark is that the page 3 is empty. Additionally I have only one question about methodology: In mostly papers proteins are identified based on 2 peptides, and the FDR is less than 1. Why in this manuscript only 1 peptide and FDR less than 4 were accepted?
Reviewer 2 Report
The manuscript addresses a fair question. The data quality is good and is well checked at various steps. However the data analysis has several flaws and needs a revision before performing other studies and drawing conclusions on the preliminary MS dataset.
I have some major concerns that I recommend authors to address before revising the paper:
Section 3.2 should come before 3.1.
Figure 1A, the DAPI in MCF-7 panel stains the whole cell. I recommend replacing this panel with a better figure. There is no scale bar in the figures. Please add.
In Figure 1B, the blots shows high abundance of actin in membrane fractions. However, membrane fraction shouldn’t contain this high amount of actin. I can understand some contaminating actin may come, but this blot is completely opposite and doesn’t justifies enriched membrane fraction. Similarly, total extract should also have some HER2, at least higher than the FT fraction.
Please increase the size of numerical values in the venn diagram. Also the text written should be darker and readable.
Using this ven diagram, I don’t understand why authors chose to focus on proteins present only in HCC1954 or MCF7 cells independently. I don’t see using FT fraction. If authors included it as negative control, then they should only include significantly up or down proteins in ME fractions and then do further GO analysis. If authors wish to compare and focus on the aggressiveness of the tumor based on cell lines, then there is no point of independent data from ME and FT fractions. You can simply compare TE for both cell lines.
Same is reflected in the GO analysis data. The proteins primarily are chromatin binding and cytoskeletal proteins. Again, the membrane protein paradigm seems of no use here.
Please explain how does the further analysis of FT fraction relevant to the scope/objectives of this study? For example Figure 2E.
Line 325: Authors say that these 155 proteins are from ME fractions only, while they are present in all four datasets. I would recommend remodifying the study design. First identify proteins significantly changes in ME as compared to FT or TE. Then perform detailed comparative analysis on HCC1954/MCF7.
Is there any significance of figure 3E. All protein classes are enriched in HCC1954. Is there more proteins having FC in HCC1954 than MCF7. In Line 336: what does FC reflects? Is it proteins in HCC1954 vx MCF7. Please explain in the text in a proper way.
Line 344: statement’ The protein PSMC3 (FC= 2.2) was responsible for the 344 proteasome increase (Figure 3D)’ is not correct. What does that mean?
Figure 4, I didn’t find any observation why authors jumped on to the cadherins and integrins? Figure 3B does not show any enrichment of cadherins in HCC1954.
Which dataset is used to make figure 3? Is it TE, ME or FT? Not clear? If it is TE, then there is no point of doing ME and FT analysis before doing the total protein analysis.
Which background lists did authors use for GO analysis in Figure 2 and 3?
Round 2
Reviewer 2 Report
The revised manuscript still lacks scientific rigor. Multiple places lack statistics on the datasets. The overall workflow is hard to follow at some places and needs a more organized way of reanalyzing the data.